# Entangled-Based Quantum Wavelength-Division-Multiplexing and Multiple-Access Networks

**DOI:** 10.3390/e25121658

**Published:** 2023-12-14

**Authors:** Marzieh Bathaee, Jawad A. Salehi

**Affiliations:** 1Sharif Quantum Center, Sharif University of Technology, Tehran 14588-89694, Iran; neda.bathaee@gmail.com; 2Institute for Convergence Science & Technology, Sharif University of Technology, Tehran 14588-89694, Iran; 3Electrical Engineering Department, Sharif University of Technology, Tehran 11155-4363, Iran

**Keywords:** entanglement, quantum network, wavelength division multiplexing

## Abstract

This paper investigates the mathematical model of the quantum wavelength-division-multiplexing (WDM) network based on the entanglement distribution with the least required wavelengths and passive devices. By adequately utilizing wavelength multiplexers, demultiplexers, and star couplers, *N* wavelengths are enough to distribute the entanglement among each pair of *N* users. Moreover, the number of devices employed is reduced by substituting a waveguide grating router for multiplexers and demultiplexers. Furthermore, this study examines implementing the BBM92 quantum key distribution in an entangled-based quantum WDM network. The proposed scheme in this paper may be applied to potential applications such as teleportation in entangled-based quantum WDM networks.

## 1. Introduction

The entangled state introduced by the EPR (Einstein, Podolsky, and Rosen) paradox in 1935 [1] threatened the completeness of quantum mechanics. Nevertheless, a violation of a Bell-type inequality based on the local hidden variables theory [2,3] confirmed the counterintuitive property that arose from the theory of quantum mechanics, e.g., non-locality [4,5]. Meanwhile, the entangled states have become one of the key quantum mechanical resources in upcoming quantum technologies, such as quantum communications [6,7], quantum computations [8], quantum teleportations [9], and quantum metrologies [10]. An extraordinary download speed via quantum teleportations [11] and unconditional security without trusting any devices [5] are some of the consequences of utilizing entanglement resources toward a futuristic quantum internet [12,13,14].

The effective distribution of these resources, entangled states, for different usages functioning as quantum communications [6,15], distributed computing [16], and so on requires reliable quantum networks [17]. Conventionally, long-haul communication needs amplifiers. Due to the no-cloning theorem arising from quantum mechanic formalism, perfect amplification is impossible in quantum networks. Instead, quantum repeaters comprising entanglement swapping [18,19] and quantum memories must be utilized for entanglement distribution and distillation over long distances. However, quantum repeaters have a long way to go to become cost-effective and reliable commercial devices. In this regard, a three-node quantum entangled-based network of remote solid-state qubits has been realized [20]. Distributing multipartite entanglement over noisy quantum networks has recently been studied in [21]. On the other hand, the local access network can distribute the entanglement among local users utilizing a broadband entangled photon source (EPS) [22,23] without relying on quantum repeaters. An untrusted entanglement provider is responsible for assigning photon pairs produced by an EPS to many subscribers. Several studies and practical realizations, such as entanglement-based QKD networks or remote state preparation (RSP) with passive and active devices, have already been studied and implemented [24,25,26,27,28,29,30,31].

Based on the strategy of quantum state sharing among all end users, any quantum home network can be categorized into two well-known classes: a prepare and measure-based network and entanglement distribution across a quantum network. While the latter is of uppermost importance in a future-proof quantum internet, it is also necessary to have loophole-free device-independent security in local networks due to imperfections in real employed devices [32]. Using the WDM scheme studied in [15], the entanglement distribution in a metropolitan optical network consists of tree-type access networks. Different experiments have been performed in the point-to-multipoint network architecture for entanglement distribution, which utilized wavelength demultiplexers and beam splitters [33,34]. Preserving non-locality in a noisy quantum network has been recently studied in the noisy intermediate-scale quantum (NISQ) era [35]. For example, ref. [36] used variational quantum optimization to investigate the criteria for distributing entanglement states and non-locality in the network using entanglement swapping. Future work can explore similar local access entangled-based WDM networks in the presence of noise, which is out of the scope of the current paper.

This paper studies entangled-based quantum WDM (QWDM) networks utilizing passive devices for access networking in the same terminology presented in [37]. We show to the best of our knowledge for the first time how, by an intelligent utilization of a star coupler and a waveguide grating router, an untrusted entanglement provider can distribute entanglement among 2N users just by selecting 2N different wavelengths from a broadband entangled source. Figure 1 illustrates the building block of our scheme for four subscribers. As depicted in Figure 1, a quantum signal composed of a wide-band entangled photon source passes through a 3×3 star coupler and gets involved in three input ports of a 4×4 waveguide grating router (WGR). Therefore, each subscriber accesses three different wavelengths out of four WGR working wavelengths. Because the four-wavelength scheme provides two distinct state pairs of entangled photons, this scheme guarantees the sharing of at least one entangled photon state between each pair of users. This work focuses on distributing entanglement in a local quantum network without utilizing entanglement swapping as experimentally investigated in previous works, such as [33,34]. This paper aims to theoretically investigate the evolution of a broadband entangled state as it passes through various WDM network components. This investigation will help to determine the necessary number of wavelengths and the rate of entanglement distribution, which are essential factors in characterizing a future quantum network.

This paper is organized as follows. Section 2 explores the entangled-based quantum WDM (QWDM) network with the least required number of wavelengths and devices. This section first studies the network utilizing a waveguide grating router (WGR) and a star coupler for a four-user network. Then, the scheme is generalized for *N* users. In Section 3, a combination of a demultiplexer and multiplexers is used in place of the WGR, and several star couplers with a lower splitting loss are utilized. Then, in Section 4, our proposed schemes are compared to the previous related works. The well-known quantum key distribution protocol BBM92 [38] as a benchmark application for entanglement versions of the quantum WDM network is presented in Section 5. Finally, the material of this paper is concluded in Section 6.

## 2. Entanglement-Based Quantum Network Based on WDM

This section studies an entanglement distribution scheme between 2N users via a combination of an (N+1)×(N+1) star coupler and a 2N×2N WGR by using only 2N different wavelengths existing in a broadband polarization-entangled photon source [22,23]. To start, we describe the entangled photon source used. Next, we investigate the entanglement of the WDM network for four and multiple users.

### 2.1. Entangled Photon Source

A broadband polarization-entangled photon source can be implemented practically and commercially by a periodically poled second-order nonlinear crystal, such as periodically poled silica fiber (PPSF) [22,23]. PPSF is well known as a compensation-free broadband entangled photon source (EPS), suitable for multi-channel quantum communications. In this structure, the entangled photon pair is generated based on the type-II spontaneous parametric down-conversion (SPDC) caused by the second-order nonlinearity response of the medium to the electric input field (pump). The second-order nonlinear term of polarization in the medium can be written as $P_{i}^{NL}=\sum \chi_{ijk}^{\left(2\right)}:E_{j}\left(\omega \right)E_{k}\left(\omega \right)$ [39], where $\chi^{\left(2\right)}$ is the second-order susceptibility. This term is responsible for creating two entangled photons called historically signal and idler after properly pumping the system, satisfying the conservation of energy ($\omega_{p}=\omega_{s}+\omega_{i}$) and the phase-matching condition ($\beta_{p}=\beta_{s}+\beta_{i}+2\pi /\Lambda$), where ω and β are the angular frequency and waveguide propagation constant, respectively, and Λ is the quasi-phase-matching period that appears whenever a periodically poled crystal is used for efficient entanglement generation via the SPDC procedure. The quantum Hamiltonian of this term is given by
(1)$$\begin{array}{c} {\hat{H}}_{\mathrm{PDC}}\propto \int_{\frac{-L}{2}}^{\frac{L}{2}}\chi^{\left(2\right)}{\hat{E}}_{p}^{\left(+\right)}\left(z,t\right){\hat{E}}_{s}^{\left(-\right)}\left(z,t\right){\hat{E}}_{i}^{\left(-\right)}\left(z,t\right)dz+h.c.,\; \end{array}$$
where
(2)$$\begin{array}{c} {\hat{E}}_{X}^{\left(+\right)}={\hat{E}}_{X}^{{\left(-\right)}^{†}}=A\sum_{m=H,V}\int d\omega_{X}\exp\left[i\left(\beta_{X}\left(\omega_{X}\right)z-\omega_{X}t\right)\right]{\hat{a}}_{m}\left(\omega_{X}\right)\;\;\;X\in \left\{p,s,i\right\},\; \end{array}$$
is the positive frequency part of the quantized electric field of the pump (X=p), signal (X=s), and idler (X=i). The parameter *A* [39] includes all the necessary constants. The length of the PPSF along the $\hat{z}$ (propagation) direction is denoted by *L*, and h.c stands for the hermitian conjugate. Furthermore, *H* and *V* represent the electric field’s horizontal and vertical polarization modes. Because the incoming pump field is strong, the pump operator can be treated as a classical field. Indeed, in the SPDC procedure, the pump field amplifies the signal and idler modes’ fluctuations in the vacuum states. Therefore, in the Schrödinger picture, the evolved state by the Hamiltonian of Equation (Equation 1) becomes
(3)$$\begin{array}{c} |\psi \left(t\right)\rangle =e^{-\frac{i}{\hbar }\int dt{\hat{H}}_{\mathrm{PDC}}\left(t\right)}|0\rangle .\; \end{array}$$
After some mathematical manipulations and keeping just the related terms [39], due to the negligible group birefringence in the PPSF [22], the biphoton output state of the EPS is expressed as
(4)$$\begin{array}{ccc} {|\psi \rangle }_{\mathrm{PDC}} & = & \frac{1}{\sqrt{2}}\int d\omega_{s}\int d\omega_{i}f\left(\omega_{s},\omega_{i}\right)\left({\hat{a}}_{H}^{†}\left(\omega_{s}\right){\hat{a}}_{V}^{†}\left(\omega_{i}\right)+{\hat{a}}_{V}^{†}\left(\omega_{s}\right){\hat{a}}_{H}^{†}\left(\omega_{i}\right)\right)|0_{s}\rangle |0_{i}\rangle \\ & = & \frac{1}{\sqrt{2}}\int d\omega_{s}\int d\omega_{i}f\left(\omega_{s},\omega_{i}\right)(|H,\omega_{s}\rangle |V,\omega_{i}\rangle +|V,\omega_{s}\rangle |H,\omega_{i}\rangle ),\; \end{array}$$
where the joint spectral amplitude (JSA), i.e., $f(\omega_{s},\omega_{i})$, has been driven by [22] as
(5)$$\begin{array}{ccc} f(\omega_{s},\omega_{i}) & \propto & \sin c(\frac{L}{2}\Delta \beta_{f}\left(\Delta \omega_{s}\right)),\\ \Delta \beta_{f} & \approx & \beta_{2}\Delta \omega_{s}^{2},\;\;\beta_{2}\approx \frac{1}{2}\left(\frac{d^{2}\beta_{H}}{d\omega^{2}}+\frac{d^{2}\beta_{V}}{d\omega^{2}}\right){|}_{\omega =\omega_{o}.}\; \end{array}$$
In Equation (Equation 5), $\Delta \omega_{s}=-\Delta \omega_{i}=\omega_{s}-\omega_{o}$, where the central spectrum is $\omega_{o}=\omega_{p}/2$. It is important to note that the signal and idler frequencies of the entangled pair are equally separated from the central spectrum $\omega_{o}$ of the JSA, $f(\omega_{s},\omega_{i})$. It is worth mentioning that the PPSF developed in [23] is coupled to a single-mode fiber, which implies that all entangled photons produced are directed in the single path (fiber), which is an important factor in utilizing this entangled source in optical fiber-based quantum communications. For instance, the star coupler based on the 2×2 fiber coupler can be easily connected to the EPS implemented by the PPSF.

### 2.2. Four-User Entanglement Distribution WDM Network Based on WGR

Figure 1 depicts our proposed scheme for entanglement distribution among four users via a waveguide grating router. A 3×3 star coupler (indicated by ${\hat{B}}^{†}$) splits a broadband entangled signal equally into three parts as inputs of a 4×4 WGR (indicated by ${\hat{A}}^{†}$). Thereby, the entangled signal produced by the EPS in stage 1 reaches stage 2 after passing the star coupler and finally distributes to users in stage 3 via the WGR as follows.


**Stage 1: Inputs**
Considering Equations (Equation 4) and (Equation 5), the input state $|\psi_{in}\rangle_{1}$ shown in Figure 1 can be rewritten according to the creation operator ${\hat{a}}_{1}^{†(1)}$, in stage 1 indicated by the superscript (1), so the input signal of the 3×3 star coupler is as follows
(6)$$\begin{array}{ccc} {|\psi \rangle }^{\left(1\right)} & = & |\psi_{in}\rangle_{1}{|0\rangle }_{2}{|0\rangle }_{3}\\ & = & \frac{1}{\sqrt{2}}\;\int \int \;d\omega_{s}d\omega_{i}f\left(\omega_{s},\omega_{i}\right)\left({\hat{a}}_{1H}^{†(1)}\left(\omega_{s}\right){\hat{a}}_{1V}^{†(1)}\left(\omega_{i}\right)+{\hat{a}}_{1V}^{†(1)}\left(\omega_{s}\right){\hat{a}}_{1H}^{†(1)}\left(\omega_{i}\right)\right)|0\rangle \\ & = & \frac{1}{\sqrt{2}}\;\int \int \;d\Delta \omega_{s}d\Delta \omega_{i}f\left(\Delta \omega_{s},\Delta \omega_{i}\right)({\hat{a}}_{1H}^{†(1)}\left(\omega_{o}+\Delta \omega_{s}\right){\hat{a}}_{1V}^{†(1)}\left(\omega_{o}+\Delta \omega_{i}\right)\\ & & \;\;\;\;\;\;\;\;\;\;\;\;\;+{\hat{a}}_{1V}^{†(1)}\left(\omega_{o}+\Delta \omega_{s}\right){\hat{a}}_{1H}^{†(1)}\left(\omega_{o}+\Delta \omega_{i}\right){)|0\rangle }_{1}{|0\rangle }_{2}{|0\rangle }_{3}\;,\; \end{array}$$
where $\omega_{s}=\omega_{o}+\Delta \omega_{s}$ and $\omega_{i}=\omega_{o}+\Delta \omega_{s}$ and subscripts *H* and *V* specify the horizontal and vertical polarizations of the signal and idler modes. The phase-matching and conservation of energy ($\omega_{s}+\omega_{i}=2\omega_{o}$) conditions imply $f(\Delta \omega_{s},\Delta \omega_{i})$ becomes dominant at $\Delta \omega_{i}=-\Delta \omega_{s}$. In the last line of Equation (Equation 6), ${|0\rangle }_{2}{|0\rangle }_{3}$ corresponds to the unused inputs of the star coupler, and subindices 1, 2, and 3 are the input port numbers of the 3×3 star coupler. Note that the first port of the star coupler receives signals containing two different mode frequencies, signal and idler, i.e., ${|0\rangle }_{1}={|0\rangle }_{s}{|0\rangle }_{i}$.
**Stage 2: Outputs of Star Coupler**
In stage 2, the mode operators are identified by the superscript (2). In this stage, the system’s state experiences a 3×3 star coupler. The 3×3 polarization-insensitive and frequency-independent star coupler ${\underline{B}}^{†}$ relates the input–output modes as
(7)$$\begin{array}{ccc} {\hat{a}}_{1P}^{\left(1\right)}\left(\omega \right) & = & B_{11}^{*}{\hat{a}}_{1P}^{\left(2\right)}\left(\omega \right)+B_{21}^{*}{\hat{a}}_{2P}^{\left(2\right)}\left(\omega \right)+B_{31}^{*}{\hat{a}}_{3P}^{\left(2\right)}\left(\omega \right),\\ {\hat{a}}_{2P}^{\left(1\right)}\left(\omega \right) & = & B_{12}^{*}{\hat{a}}_{1P}^{\left(2\right)}\left(\omega \right)+B_{22}^{*}{\hat{a}}_{2P}^{\left(2\right)}\left(\omega \right)+B_{32}^{*}{\hat{a}}_{3P}^{\left(2\right)}\left(\omega \right),\\ {\hat{a}}_{3P}^{\left(1\right)}\left(\omega \right) & = & B_{13}^{*}{\hat{a}}_{1P}^{\left(2\right)}\left(\omega \right)+B_{23}^{*}{\hat{a}}_{2P}^{\left(2\right)}\left(\omega \right)+B_{33}^{*}{\hat{a}}_{3P}^{\left(2\right)}\left(\omega \right),\; \end{array}$$
where P∈{H,V}. Star couplers can be implemented using several 2×2 fiber couplers or beam splitters and/or formed using the fused biconical tapering method, as discussed in Chapter 6 of [40]. Due to energy conservation, the coefficients that relate the output signals to the input signals construct a unitary transfer matrix $\underline{B}$. The coefficient $B_{ij}$, the matrix element of $\underline{B}$ that connects input *i* to output *j*, is derived from the transmission and reflection coefficients of the used fiber couplers or beam splitters. References such as [41,42] studied the star coupler in the context of quantum mechanics, where the relation between the annihilation operator of the input modes and output modes is written similarly to Equation (Equation 7). In an n×n balanced star coupler, the input field is equally divided and transmitted to each output. Hence, $|B_{ij}{|}^{2}={\left|B_{n}\right|}^{2}=1/n$ for all *i* and *j*. By substituting Equation (Equation 7) in Equation (Equation 6), the evolved input state at the inputs of the WGR (or, equivalently, the state ${|\psi \rangle }^{\left(1\right)}$ in the bases of the output modes) becomes
(8)$$\begin{array}{ccc} {|\psi \rangle }^{\left(2\right)} & = & \frac{1}{\sqrt{2}}\;\int \int \;d\Delta \omega_{s}d\Delta \omega_{i}f\left(\Delta \omega_{s},\Delta \omega_{i}\right)\sum_{l,m=1}^{3}B_{l1}B_{m1}\\ & & \times \left[{\hat{a}}_{lH}^{†(2)}\left(\omega_{s}\right){\hat{a}}_{mV}^{†(2)}\left(\omega_{i}\right)+{\hat{a}}_{lV}^{†(2)}\left(\omega_{s}\right){\hat{a}}_{mH}^{†(2)}\left(\omega_{i}\right)\right]{|0\rangle }_{1}{|0\rangle }_{2}{|0\rangle }_{3}.\; \end{array}$$To better understand the different summation terms in Equation (Equation 8), we provide its expanded form in Equation (Equation 18) of Appendix A.
**Stage 3: Outputs of WGR**
The lossless 4×4 WGR input–output mode relations can be summarized in a frequency-dependent matrix form
(9)$$\begin{array}{ccc} & & \left[\begin{array}{c} {\hat{a}}_{1P}^{\left(2\right)}\left(\omega \right)\\ {\hat{a}}_{2P}^{\left(2\right)}\left(\omega \right)\\ {\hat{a}}_{3P}^{\left(2\right)}\left(\omega \right)\\ {\hat{a}}_{4P}^{\left(2\right)}\left(\omega \right)\; \end{array}\right]={\underline{A}}^{†}\left[\begin{array}{c} {\hat{a}}_{1P}^{\left(3\right)}\left(\omega \right)\\ {\hat{a}}_{2P}^{\left(3\right)}\left(\omega \right)\\ {\hat{a}}_{3P}^{\left(3\right)}\left(\omega \right)\\ {\hat{a}}_{4P}^{\left(3\right)}\left(\omega \right)\; \end{array}\right],\; \end{array}$$
where
(10)$$\begin{array}{ccc} {\underline{A}}^{†} & = & \left[\begin{array}{cccc} A_{11}^{*}\left(\omega \right) & A_{21}^{*}\left(\omega \right) & A_{31}^{*}\left(\omega \right) & A_{41}^{*}\left(\omega \right)\\ A_{12}^{*}\left(\omega \right) & A_{22}^{*}\left(\omega \right) & A_{32}^{*}\left(\omega \right) & A_{42}\left(\omega \right)\\ A_{13}^{*}\left(\omega \right) & A_{23}^{*}\left(\omega \right) & A_{33}^{*}\left(\omega \right) & A_{43}^{*}\left(\omega \right)\\ A_{14}^{*}\left(\omega \right) & A_{24}^{*}\left(\omega \right) & A_{34}^{*}\left(\omega \right) & A_{44}^{*}\left(\omega \right)\; \end{array}\right].\; \end{array}$$For an ideal WGR, it is assumed
(11)$$\begin{array}{c} |A_{ij}^{*}{\left(\omega \right)|}^{2}\approx \left\{\begin{array}{cc} 1 & |\omega -\omega_{ij}|\leq \delta \omega /2\\ 0 & \mathrm{otherwise} \end{array}\right., \end{array}$$
where $\omega_{ji}$ is the central frequency of the passing bands for any pair of the input–output of the WGR with bandwidth δω [43,44]. The distinguished central frequencies passing through the 4×4 WGR from its input *j* toward its output *i* are indicated by $\omega_{ij}=\omega_{l}$, where l=i−j+1mod4 and i,j∈{1,2,3,4} [40]. Whenever i−j+1=4 or i−j+1=0, we assign l=4. So, by choosing two signal frequencies ($\omega_{s_{1}}$ and $\omega_{s_{2}}$) and two idler frequencies ($\omega_{i_{1}}$ and $\omega_{i_{2}}$), which are set on the WGR-guided frequencies, as follows
(12)$$\begin{array}{ccc} \omega_{11} & = & \omega_{22}=\omega_{33}=\omega_{44}=\omega_{s_{1}}=\frac{2\pi c}{\lambda_{1}},\\ \omega_{21} & = & \omega_{32}=\omega_{43}=\omega_{14}=\omega_{s_{2}}=\frac{2\pi c}{\lambda_{2}},\\ \omega_{41} & = & \omega_{12}=\omega_{23}=\omega_{34}=\omega_{i_{1}}=\frac{2\pi c}{\lambda_{4}},\\ \omega_{31} & = & \omega_{42}=\omega_{13}=\omega_{24}=\omega_{i_{2}}=\frac{2\pi c}{\lambda_{3}},\; \end{array}$$
one can approximate that the nonzero and dominant terms of Equation (Equation 9) occur at the WGR-passing frequencies as
(13)$$\begin{array}{ccc} {\hat{a}}_{jP}^{\left(2\right)}\left(\omega \right) & \approx & \left\{\begin{array}{cc} A_{ij}^{*}\left(\omega \right){\hat{a}}_{iP}^{\left(3\right)}\left(\omega \right) & |\omega -\omega_{ij}|\leq \delta \omega /2,\;\forall i\in \left\{1,2,3,4\right\}\\ 0 & \mathrm{otherwise} \end{array}\right..\; \end{array}$$In stage 3, the WGR acts as a frequency filter with the bandwidth δω of each output, and the passing signals satisfy the condition $\Delta \omega_{s}=-\Delta \omega_{i}$. Due to the phase-matching and energy-conservation conditions, the spectral joint amplitudes are $f(\Delta \omega_{s_{1}},\Delta \omega_{i_{2}})\approx 0$ and $f(\Delta \omega_{s_{2}},\Delta \omega_{i_{1}})\approx 0$. Thus, the nonzero terms $f(\Delta \omega_{s_{1}},\Delta \omega_{i_{1}})$ and $f(\Delta \omega_{s_{2}},\Delta \omega_{i_{2}})$, indicated by $f(\Delta \omega_{s_{1}})$ and $f(\Delta \omega_{s_{2}})$ from now on, can only be contributed in the following calculations. The spectral profiles of the WGR outputs are assumed to be a square with a width of δω. Therefore, we apply the following approximation in the integration terms that appear after inserting Equation (Equation 13) in Equation (Equation 8).
(14)$$\begin{array}{c} \int_{\Delta \omega_{s_{1(2)}}-\delta \omega /2}^{\Delta \omega_{s_{1(2)}}+\delta \omega /2}\int_{\Delta \omega_{i_{1(2)}}-\delta \omega /2}^{\Delta \omega_{i_{1(2)}}+\delta \omega /2}d\Delta \omega_{s}d\Delta \omega_{i}f\left(\Delta \omega_{s},\Delta \omega_{i}\right)A_{j_{1}l}\left(\omega_{s}\right)A_{j_{2}m}\left(\omega_{i}\right){\hat{a}}_{j_{1}P}^{†(3)}\left(\omega_{s}\right){\hat{a}}_{j_{2}P^{\prime }}^{†(3)}\left(\omega_{i}\right)\\ \approx \delta \omega f\left(\Delta \omega_{s_{1(2)}}\right)A_{j_{1}l}\left(\omega_{s_{1(2)}}\right)A_{j_{2}m}\left(\omega_{i_{1(2)}}\right){\hat{a}}_{j_{1}P,s_{1(2)}}^{†(3)}{\hat{a}}_{j_{2}P^{\prime },i_{1(2)}}^{†(3)},\; \end{array}$$
where
(15)$$\begin{array}{c} {\hat{a}}_{j_{1}P,s_{1(2)}}^{†(3)}\approx \int_{\Delta \omega_{s_{1(2)}}-\delta \omega /2}^{\Delta \omega_{s_{1(2)}}+\delta \omega /2}d\Delta \omega_{s}{\hat{a}}_{j_{1}P}^{†(3)}\left(\omega_{s}\right)/\sqrt{\delta \omega },\\ {\hat{a}}_{j_{2}P^{\prime },i_{1(2)}}^{†(3)}\approx \int_{\Delta \omega_{i_{1(2)}}-\delta \omega /2}^{\Delta \omega_{i_{1(2)}}+\delta \omega /2}d\Delta \omega_{i}{\hat{a}}_{j_{2}P^{\prime }}^{†(3)}\left(\omega_{i}\right)/\sqrt{\delta \omega }.\; \end{array}$$In Equation (Equation 14), $P,P^{\prime }\in \left\{H,V\right\}$, l,m∈{1,2,3}, and $j_{1},j_{2}\in \left\{1,2,3,4\right\}$. Moreover, according to Equations (Equation 12) and (Equation 13), for integral bounds around $\omega_{s_{1}}$ and $\omega_{i_{1}}$, $j_{1}=l$ and $j_{2}=3+m\;\mathrm{mod}\;4$, and for integral bounds around $\omega_{s_{2}}$ and $\omega_{i_{2}}$, $j_{1}=l+1\;\mathrm{mod}\;4$ and $j_{2}=m+2\;\mathrm{mod}\;4$. Note that whenever $j_{1}$ and $j_{2}$ become zero, we assign them 4.Inserting Equation (Equation 9) in Equation (Equation 8) and using Equations (Equation 12)–(Equation 15), the input state is rewritten based on the representation of the modes in stage 3 as
(16)$$\begin{array}{ccc} & & {|\psi \rangle }^{\left(3\right)}=\frac{\delta \omega f(\Delta \omega_{s_{1}})}{\sqrt{2}}\sum_{l,m=1}^{3}B_{l1}B_{m1}A_{ll}\left(\omega_{s_{1}}\right)A_{3+m\;\mathrm{mod}\;4,m}\left(\omega_{i_{1}}\right)\\ & & \;\;\;\;\;\;\;\;\;\;\;\;\;\;\;\;\;\;\;\;\;\;\;\;\;\;[{\hat{a}}_{lH,s_{1}}^{†(3)}{\hat{a}}_{3+m\;\mathrm{mod}\;4,V,i_{1}}^{†(3)}+{\hat{a}}_{lV,s_{1}}^{†(3)}{\hat{a}}_{3+m\;\mathrm{mod}\;4,H,i_{1}}^{†(3)}]\\ & & \;\;\;\;\;\;\;\;\;\;\;\;\;\;\;\;\;\;\;\;\;\;\;\;\;\;\;\;\;\;\;\;\;\;\;\;\;\;\;\;\;\;\times {|0\rangle }_{1}{|0\rangle }_{2}{|0\rangle }_{3}{|0\rangle }_{4}\\ & & \;\;\;\;\;\;\;\;\;\;\;\;+\frac{\delta \omega f(\Delta \omega_{s_{2}})}{\sqrt{2}}\sum_{l,m=1}^{3}B_{l1}B_{m1}A_{l+1\;\mathrm{mod}\;4,l,s_{2}}A_{m+2\;\mathrm{mod}\;4,m,i_{2}}\\ & & \;\;\;\;\;\;\;\;\;\;\;\;\;\;\;\;\;\;\;\;\;\;\;\;\;\;\;[{\hat{a}}_{l+1\;\mathrm{mod}\;4,H,s_{2}}^{†(3)}{\hat{a}}_{m+2\;\mathrm{mod}\;4,V,i_{2}}^{†(3)}+{\hat{a}}_{l+1\;\mathrm{mod}\;4,V,s_{2}}^{†(3)}{\hat{a}}_{m+2\;\mathrm{mod}\;4,H,i_{2}}^{†(3)}]\\ & & \;\;\;\;\;\;\;\;\;\;\;\;\;\;\;\;\;\;\;\;\;\;\;\;\;\;\;\;\;\;\;\;\;\;\;\;\;\;\;\;\;\times {|0\rangle }_{1}{|0\rangle }_{2}{|0\rangle }_{3}{|0\rangle }_{4},\; \end{array}$$Whenever the subindex in Equation (Equation 16) becomes a multiple of 4, we assign that label to 4 instead of 0. As a result, in Equation (Equation 16), each of the four users is indicated from the mode subindex set {1,2,3,4}.The first three lines on the right-hand side of Equation (Equation 16) are related to the photon entangled pair $\lambda_{1}$ and $\lambda_{4}$ ($\omega_{s_{1}}$ and $\omega_{i_{1}}$), which is distributed between all the users. The second set of three lines in Equation (Equation 16) describes the entangled photon pair $\lambda_{2}$ and $\lambda_{3}$ ($\omega_{s_{2}}$ and $\omega_{i_{2}}$) shared among all the users. In Appendix A, Equation (Equation 16) is expanded on the *m* and *l* indices, and the explicit form of the entangled state between each pair of users is specified in Equation (Equation 20) (see also Figure 1). Therefore, from Equations (Equation 16) and (Equation 20), it is clear that there are at least two entangled states between any two users. Furthermore, the pairs specified by subscripts (1,2) and (3,4) share three entangled states (see Equation (Equation 20)). As a result, the fully connected quantum network is realized using only four wavelengths with a minimum number of utilized devices, i.e., a star coupler preceded by a waveguide grating router. This scheme paves the way for the miniaturization of a future quantum network.

### 2.3. Generalized Entanglement Distribution WDM Network Based on WGR for 2N Users

A generalized model of entanglement distribution for 2N users is illustrated in Figure 2. This generalized scheme inserts a broadband polarization-entangled photon source into an (N+1)×(N+1) star coupler from its first input port. Subsequently, a WGR distributes N+1 out of 2N different wavelengths in the EPS spectrum to each subscriber, as depicted on the right-hand side of Figure 2. Consider the *i*th user with the wavelength set as
$$\begin{array}{c} \Lambda_{i}=\left(\lambda_{i},\lambda_{i-1\;\mathrm{mod}\;2N},\cdots ,\lambda_{i-l\;\mathrm{mod}\;2N},\cdots ,\lambda_{i-N\;\mathrm{mod}\;2N}\right).\; \end{array}$$
The modes related to the above wavelength set are entangled one by one, with the modes corresponding to the following wavelength set as
$$\begin{array}{c} \Lambda_{i}^{*}=\left(\lambda_{2N-i+1\;\mathrm{mod}\;2N},\lambda_{2N-i+2\;\mathrm{mod}\;2N},\cdots ,\lambda_{2N-i+l\;\mathrm{mod}\;2N},\cdots ,\lambda_{2N-i+N+1\;\mathrm{mod}\;2N}\right),\; \end{array}$$
which is called a conjugate wavelength set. The cardinal number of the sets $\Lambda_{i}$ and $\Lambda_{i}^{*}$ equals N+1, while the total number of wavelengths is 2N. Because $\Lambda_{i}\cup \Lambda_{j}^{*}\subseteq \left(\lambda_{1},\cdots ,\lambda_{2N}\right)$, the cardinal number of $\Lambda_{i}\cup \Lambda_{j}^{*}$ is equal or less than 2N, i.e., $N+1\leq |\Lambda_{i}\cup \Lambda_{j}^{*}|\leq 2N$. Therefore, $\Lambda_{i}^{*}\cap \Lambda_{j}\neq ⌀\;\forall i,j$ means at least one entangled pair exists between the *i*th and *j*th users. Note that whenever imod2N becomes 0, we assign that label the value 2N (i.e., $\lambda_{0}=\lambda_{2N}$).

The chief drawback of this scheme is related to the degradation of the entanglement distribution rate with the factor $1/{\left(N+1\right)}^{2}$ introduced by the (N+1)×(N+1) star coupler. The scheme presented in the next section improves this factor to $1/{\left(N\right)}^{2}$ (if we neglect the loss of the multiplexers). However, a salient trade-off still exists between the number of required wavelengths for running the fully connected network and the entanglement distribution rate of each pair in the network.

## 3. Entanglement-Based Quantum WDM Network Based on Wavelength Demultiplexing and Multiplexing

Here, we study a different configuration for distributing entanglement between end users in a quantum network. Figure 3 and Figure 4 illustrate four- and eight-user entangled networks, respectively. Accordingly, the generalized setup shown in Figure 5 utilizes a 1×2N wavelength demultiplexer and the *N* number of N×1 wavelength multiplexers. These wavelength multiplexers merge different wavelengths in a single fiber to send to the related long-distance user. Moreover, the *N* number of N×N balanced star couplers are deployed in each demultiplexer output. Substituting an (N+1)×(N+1) balanced beam splitter with N×N balanced star couplers in this new scheme leads to an entanglement rate increase, provided the loss arising from the multiplexer increment is neglected. The global state at the user’s premises can be derived easily in a similar manner performed in the previous section. It is enough to write the input state $|\psi_{in}\rangle$ based on the output modes at the users’ sites. For example, for the four-user network depicted in Figure 3, first, the input modes of the EPS evolve through the 1×4 demultiplexer according to Equation (Equation 9) (its quantum model is the same as the 4×4 WGR with the three unused input ports). Then, each output mode of the demultiplexer passes through a 2×2 beam splitter where the output modes are related to the input modes akin to Equation (Equation 7) but for a 2×2 system. Finally, two output modes related to two different beam splitters’ outputs, as depicted in Figure 3, are combined by a 2×1 wavelength multiplexer. The relation between the input and output modes of the 2×2 wavelength multiplexer is the same as a 2×2 WGR. In Figure 3, for the sake of simplicity, all the 2×1 wavelength multiplexers are indicated by the same operator ${\hat{A}}^{†}$, and the input and output ordering of their ports are chosen arbitrarily so that the desirable multiplexing happens. Indeed, we assume that all 2×1 wavelength multiplexers operating at their specified two wavelengths are constructed such that their ${\hat{A}}^{†}$ operator has the same matrix elements, regardless of the two wavelengths chosen to operate for them.

To prove why this scheme gives rise to a fully connected entanglement network, consider the *i*th user with the wavelength set as
$$\begin{array}{ccc} \Lambda_{i} & = & (\lambda_{i},\lambda_{i+1\;\mathrm{mod}\;2N},\cdots ,\lambda_{i+l\;\mathrm{mod}\;2N},\cdots ,\lambda_{i+N-1\;\mathrm{mod}\;2N}).\; \end{array}$$
Here, we assume the modes related to the above wavelength set are entangled one by one, with the modes corresponding to the following wavelength set as
$$\begin{array}{ccc} & & \Lambda_{i}^{*}=\left(\lambda_{i+N\;\mathrm{mod}\;2N},\lambda_{i+1+N\;\mathrm{mod}\;2N},\cdots ,\lambda_{i+l+N\;\mathrm{mod}\;2N},\cdots ,\lambda_{i+N-1+N\;\mathrm{mod}\;2N}\right),\; \end{array}$$
which is called a conjugate wavelength set. It is important to note that $\Lambda_{i}\cap \Lambda_{i}^{*}=⌀$ and $\Lambda_{i}\cup \Lambda_{i}^{*}=\left(\lambda_{1},\cdots ,\lambda_{2N}\right)$. It is straightforward to show that at least one common wavelength exists between each conjugate wavelength set (one user) and other wavelength sets (other users). To achieve this, one can employ the method of proof by contradiction. Consider two different users with index *i* and *j*. If there is no common wavelength between sets $\Lambda_{i}^{*}$ and $\Lambda_{j}$, one deduces i=j because the total number of used wavelengths is 2N. As a result, this configuration leads to entanglement distribution between every pair.

## 4. Comparison with Previous Realizations

This section will compare this paper’s results with previous works that used passive optical WDM communication components. This comparison will focus on the number of wavelengths and devices used and the rate of entanglement distribution between users. This information is summarized in Table 1. Additionally, to provide a clearer comparison, we use similar block diagrams of the devices presented in this paper to illustrate the setups of the two previous experiments.

To compare our setup with the previous works [33,34], we depict the schematic configurations of these experimental setups based on the WDM demultiplexer/multiplexers and beam splitters, respectively, in Figure 6 and Figure 7. Likewise, any entangled state compromises two single photons labeled by the wavelength pairs $\left(\lambda_{i},\lambda_{N-i+1}\right)\;\forall i\in \left\{1,\cdots ,N/2\right\}$, which correspond to the signal and idler modes, respectively. It is assumed that the number of users, *N*, is even.

The setup presented in Figure 6 utilized twelve wavelengths to have a fully connected logical network layer among four users, while we used four wavelengths in our scheme presented in Figure 1. Nevertheless, it is essential to mention that because wavelength multiplexers substitute a star coupler in the setup related to Figure 6, the coincident photon count between any pair becomes proportional to ${\left|A\right|}^{8}$ in comparison to our model, which is proportional to ${2|A|}^{4}{\left|B_{3}\right|}^{4}$ (see Equation (Equation 16)). The coefficient 2 appearing in the coincident photon count shows that at least two entangled states are shared between each pair in our model. Moreover, we assume $|A_{ij}\left|=|A\right|$ and $|B_{ij}\left|=\right|B_{3}|=1/\sqrt{3}$ for all *i* and *j*. Suppose the loss corresponding to the multiplexers/demultiplexers is less than the loss introduced by the 3×3 star coupler in our model (roughly 4.8dB). In that case, the entanglement distribution rate of our model falls behind [33]. However, our scheme becomes dominant with respect to the number of required wavelengths when the number of users increases. For example, for eight-user networks, [33] needs 56 wavelengths, whereas our scheme utilizes only eight wavelengths. Note that Figure 1 can introduce a fully connected entangled network even by sharing two wavelengths, $\lambda_{1}$ and $\lambda_{4}$, or $\lambda_{2}$ and $\lambda_{3}$.

In 2020, [34] demonstrated that an eight-user entanglement-based quantum network requires only 16 wavelengths to become fully connected by utilizing beam splitters at the outputs of the wavelength demultiplexer, which separates the wavelength contents of the broadband entangled source. However, based on the generalized form of our scheme explained in Section 2.3, our proposed eight-user quantum network requires only eight wavelengths if a star coupler is placed before a WGR. Figure 3 illustrates the four-user versions of the quantum network used by [34]. Hence, it is comparable with our four-user model now. In both models, the fully connected network runs only via four wavelengths.

Table 1 gives the scales of the entanglement distribution rate for different entangled-based network configurations. To obtain this scale, consider the entangled states containing two photons passing through each network device. Hence, the matrix elements of the network device appear two times as a coefficient of the entangled state (see Equations (Equation 16) and (Equation 20)–(Equation 24)). If the network has $m=m_{1}+m_{2}+m_{3}+m_{4}$ WDM components, such as $m_{1}$ numbers of n×n balanced star couplers, $m_{2}$ wavelength multiplexers, $m_{3}$ wavelength demultiplexers, and $m_{4}$ WGRs, the coefficient of each shared entangled state is proportional to $|B_{n}{|}^{m_{1}}{\left|A\right|}^{m_{2}+m_{3}+m_{4}}$. Therefore, the probability of each entangled state produced is proportional to $|B_{n}{|}^{2m_{1}}{\left|A\right|}^{2(m_{2}+m_{3}+m_{4})}$, which estimates the entanglement distribution rate. For simplicity, all the matrix elements in the wavelength multiplexers, demultiplexers, and WGRs with different numbers of ports are indicated by |A|.

## 5. Entanglement-Based BBM92 QKD Protocol Used in the WDM Network

The BBM92 QKD protocol [38] is the entangled version of the BB84 QKD protocol. In this protocol, an entangled bipartite state such as $|\psi_{in}\rangle =\frac{1}{\sqrt{2}}(|HV\rangle +\left|VH\rangle \right)=\frac{1}{\sqrt{2}}(|DD\rangle -\left|AA\rangle \right)$ is distributed between two legitimate users by an untrusted entanglement provider, where $|m\rangle ={\hat{a}}_{m}^{†}|0\rangle$ for the polarization modes m∈{H,V,D,A} and $|D\rangle =\frac{1}{\sqrt{2}}(|H\rangle +|V\rangle ),\;|A\rangle =\frac{1}{\sqrt{2}}(|H\rangle -\left|V\rangle \right)$. The users randomly choose rectilinear or diagonal bases for measuring the polarization of their received photon. As it is clear from the form of $|\psi_{in}\rangle$, if both users choose the rectilinear (diagonal) basis, their achieved information is anti-correlated (correlated). Like the BB84 QKD protocol, *H* and *D* polarization are related to the classical bit 0, while 1 is encoded by *V* or *A* polarization.

To understand how BBM92 is an entangled version of BB84, we can assume that Alice uses an entangled photon source as a transmitter in the BB84 protocol. She randomly measures the polarization of one photon of the bipartite system from a rectilinear or diagonal basis and sends the other to Bob. This procedure is equivalent to a random selection of the photon polarization via the polarization modulator. Because it can be assumed that Alice does not reveal the result of her measurement to herself until Bob performs the measurement, the BB84 protocol mimics the BBM92 protocol, where the entanglement provider is at the transmitter’s site.

As depicted in Figure 8, a four-user entangled-based network consists of an entanglement provider as well as four users, Alice (A), Bob (B), Charlie (C), and David (D), equipped with measurement facilities. The entanglement provider distributes to four users a polarization-based bipartite entangled state produced by an entangled photon source (EPS), such as a PPSF, via a 2×2 star coupler and a WGR. Photon pairs with wavelengths ($\lambda_{1}$ and $\lambda_{4}$) as well as ($\lambda_{2}$ and $\lambda_{3}$) are entangled in their polarization degrees according to Equation (Equation 4). The table in Figure 8 displays the wavelength allocated to each user. For instance, with the help of Equations (Equation 16) and (Equation 20), with the probability $P\propto {\left(|B\right|}^{2}{\left|A\right|}^{2}{)}^{2}$, the entangled state shared between Alice and Charlie can be written as
(17)$$\begin{array}{ccc} {|\psi \rangle }_{AC}= & & \frac{1}{2}{[\left({\hat{a}}_{A,H,\lambda_{4}}^{†}{\hat{a}}_{C,V,\lambda_{1}}^{†}+{\hat{a}}_{A,V,\lambda_{4}}^{†}{\hat{a}}_{C,H,\lambda_{1}}^{†}\right)|00\rangle }_{AC}\\ & & \;\;\;\;\;+\left({\hat{a}}_{A,H,\lambda_{3}}^{†}{\hat{a}}_{C,V,\lambda_{2}}^{†}+{\hat{a}}_{A,V,\lambda_{3}}^{†}{\hat{a}}_{C,H,\lambda_{2}}^{†}\right){|00\rangle }_{AC}],\; \end{array}$$
where the creation operator ${\hat{a}}_{i,p,\lambda_{j}}^{†}$ is related to the photon in the site of i∈{A,C} with the polarization p∈{H,V} around the central wavelength $\lambda_{j}$ where j∈{1,2,3,4}. According to Equation (Equation 17), two entangled states are shared between Alice and Charlie in this network, which could also be understood by the fact that the common elements of the conjugate wavelength set of Alice’s wavelength set $\left\{\lambda_{1},\lambda_{3},\lambda_{4}\right\}$ (i.e., $\left\{\lambda_{4},\lambda_{2},\lambda_{1}\right\}$) with Charlie’s wavelength set $\left\{\lambda_{1},\lambda_{2},\lambda_{3}\right\}$ are $\left\{\lambda_{1},\lambda_{2}\right\}$. Due to the fact that it is assumed that, in each mode, there is a single photon, photon coincidence counting happens probabilistically for only one pair at a time. Consequently, the secret key shared between pairs is unique.

## 6. Conclusions

The authors of this paper studied the WDM network’s entangled-based quantum state sharing between users by minimizing the required wavelengths for a fully connected entangled network. It was shown that each receiver could receive the encoded information in the quantum signal from every transmitter. Still, two receivers cannot receive the same data from the same transmitter at a specific time. In the entangled-based network, we proved that using an N×N star coupler behind a 2N×2N WGR, the minimum number of required wavelengths for distributing entanglement among each pair in the quantum WDM network becomes 2N. The previous related entangled WDM networks were compared with our scheme. This comparison highlights the advantage of our work in requiring fewer wavelengths and devices, which is crucial in developing a cost-effective and compact quantum local access network. Finally, we concluded the paper by showing how the QKD protocol, BBM92, was employed in the WDM network.

## Figures and Tables

**Figure 1 entropy-25-01658-f001:**
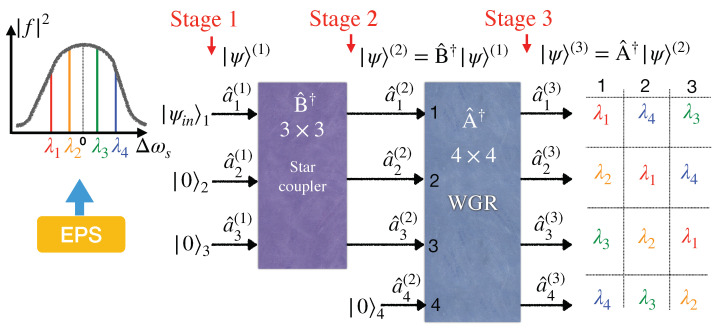
Schematic configuration of four quantum entanglement distribution network users. EPS is an entangled photon source. WGR stands for waveguide grating router.

**Figure 2 entropy-25-01658-f002:**
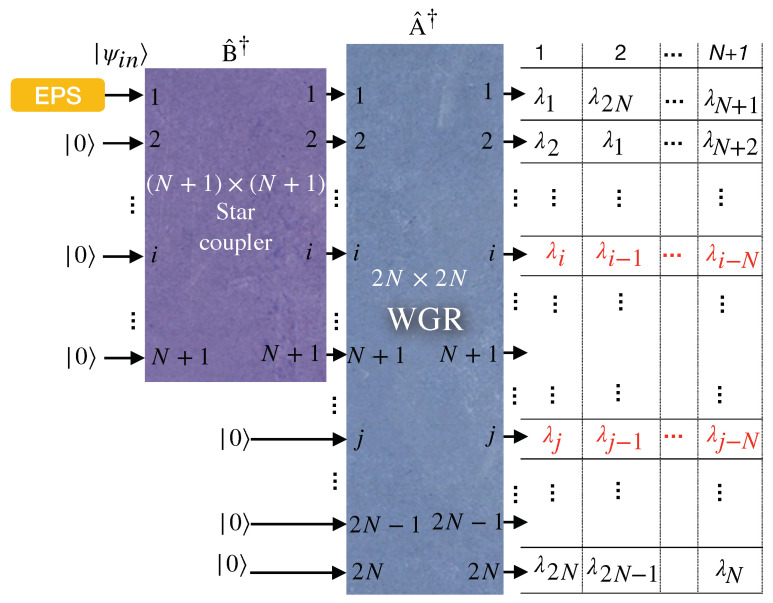
Schematic configuration of a quantum entanglement distribution network for 2N users.

**Figure 3 entropy-25-01658-f003:**
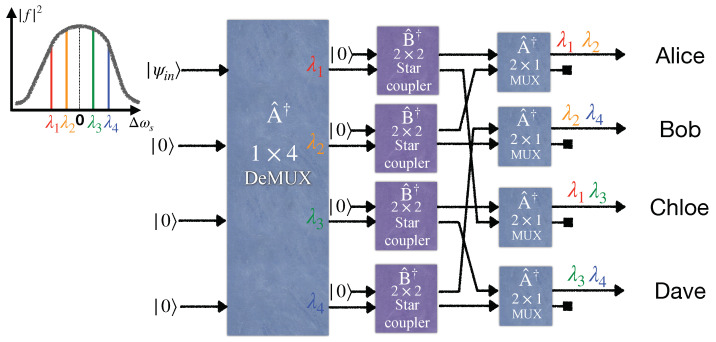
An entanglement distribution network between four end users via broadband entangled photon sources utilizing a wavelength demultiplexer (DeMUX), four 2×2 beam splitters, and four multiplexers (MUX).

**Figure 4 entropy-25-01658-f004:**
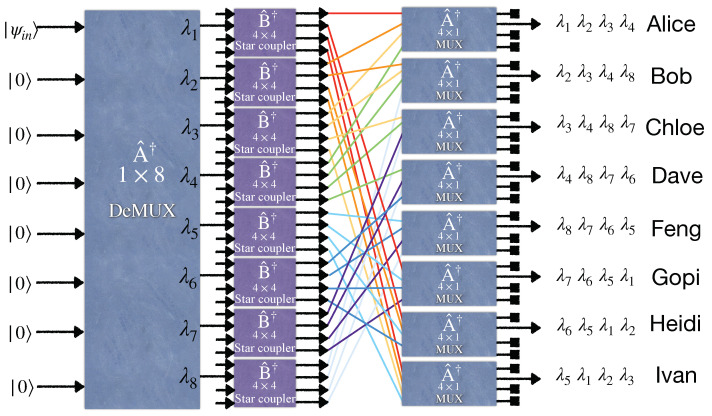
An entanglement distribution network between eight end users via broadband entangled photon sources utilizing a wavelength demultiplexer, eight 4×4 beam splitters, and eight multiplexers.

**Figure 5 entropy-25-01658-f005:**
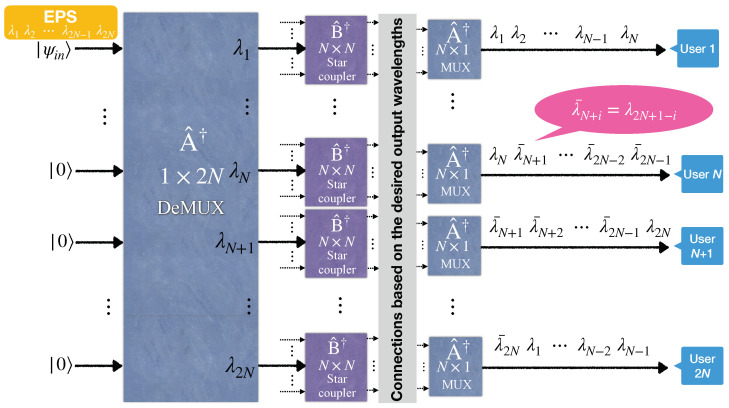
Schematic configuration of a quantum entanglement distribution network for 2N users based on wavelength DeMUX (demultiplexer) and MUX (multiplexer).

**Figure 6 entropy-25-01658-f006:**
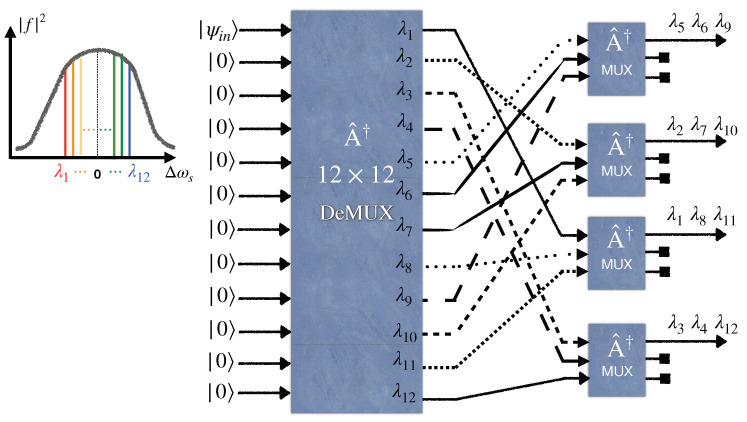
Schematic configuration of a quantum entanglement distribution network for 4 users according to [33].

**Figure 7 entropy-25-01658-f007:**
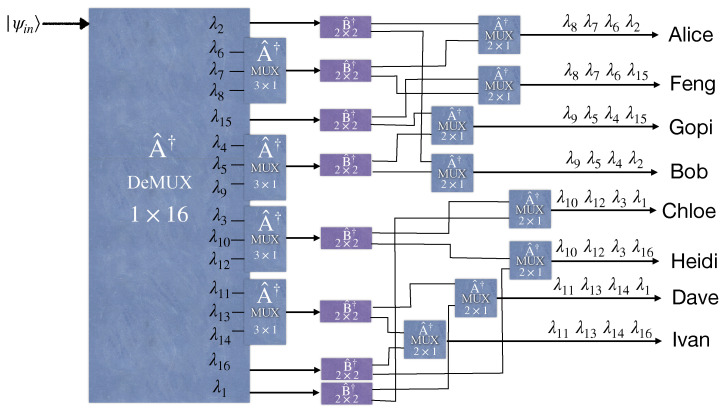
Schematic configuration of a quantum entanglement distribution network for eight users according to [34].

**Figure 8 entropy-25-01658-f008:**
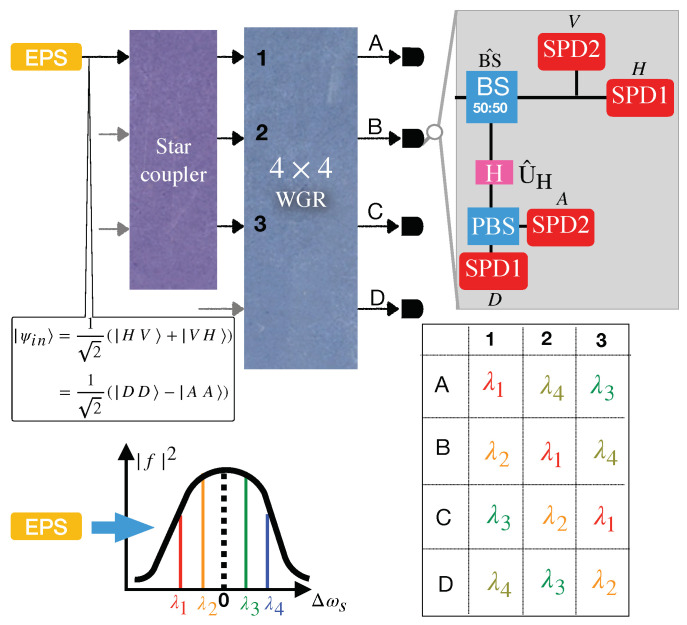
A four-user BBM92 QKD entangled-based network. The measurement components are BS (beam splitter), PBS (polarization beam splitter), SPD (single photon detector), and H (half-wave plate). EPS is an entangled photon source, and WGR is a waveguide grating router.

**Table 1 entropy-25-01658-t001:** Comparison of the entangled-based QWDM networks presented in this paper and the references [33,34]. The number of users is specified by *N*. Because ideal multiplexers, demultiplexers, and WGRs have the same mathematical model, their matrix elements’ value is assumed to be equal and represented by |A|. The matrix elements’ value of the n×n balanced star coupler is indicated by $|B_{n}|=\frac{1}{\sqrt{n}}$.

N-User Entangled-Based Network	Number of Required Wavelengths	Number of Employed Devices	Scale of Entanglement Distribution’s Rate
Network presented in Figure 1: N = 4(8)	4(8)	One 3×3 (7×7) star coupler and one 4×4 (8×8) WGR	$2|B_{3(7)}{|}^{4}{\left|A\right|}^{4}$
Network presented in Figure 3 and Figure 4: N = 4(8)	4(8)	Four (eight) 2×2 (4×4) star couplers, one 1×4 (1×8) demultiplexer, and four (eight) 2×1 (4×1) multiplexers	$|B_{2(4)}{|}^{4}{\left|A\right|}^{8}$
Ref. [33]: N = 4(8)	12(56)	One 1×12 (1×56) demultiplexer, and four (eight) 3×1 (7×1) multiplexers	${\left|A\right|}^{8}$
Ref. [34]: N = 4(8)	4(16)	Four (eight) 2×2 star couplers, one 1×4 (1×16) demultiplexer, four (eight) 2×1 multiplexers, and zero (four) 3×1 multiplexers	$|B_{2}{|}^{4}{\left|A\right|}^{8(12)}$

## Data Availability

Data is contained within the article.

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
