# Peer review of "Entangled-Based Quantum Wavelength-Division-Multiplexing and Multiple-Access Networks"

_entropy, 2023, doi:10.3390/e25121658_

Round 1

Reviewer 1 Report

Comments and Suggestions for Authors

The results, presented in the manuscript entropy-2695253, are interesting and valid enough to be published. The results are cleary presented. Moreover, the authors provide a sufficient list of publications introducing the reader to the discussed topics. Nevertheless, in my opinion, a minor revision of this article is required.

In the conclusion, the authors write 'The previous related entangled WDM networks were compared with our scheme.' In my opinion, this sentence should be expanded.

Authors should read the entire text carefully and correct language errors and typos, for example:

Lines 241 and 247: BBM --> BBM92
Line 293: einstein podolsky rosen --> Einstein Podolsky Rosen

Comments on the Quality of English Language

Authors should correct language errors and typos.

Reviewer 2 Report

Comments and Suggestions for Authors

The manuscript theoretically investigates the entanglement distribution for 2N users by using WDMs and star couplers, and authors provide a perspective to the QKD network by theoretically demonstrating a BBM92 protocol. I found that the results can provide an interesting and useful direction for quantum network. It seems to be suitable for publication in Entropy after addressing several comments raised in this report.

1.      In Eq. (6), it seems that there are four input ports, but, in Figure 1, the 3 x 3 star coupler is considered. It can make readers misunderstood. I recommend authors to rearrange the clear statements and equations with the experimental conditions to ensure consistency.

2.      At the stage 2 (line 119), there is no explanation about the coefficient B. The coefficient B can have the amplitude and relative phase between the modes. The related contents are in PHYSICALREVIEWA68,052315 (2003). I believe that the coefficient of B must be explained somewhere in the main text.

3.      The Equation (14) provides a meaningful and important results in the miniaturization of quantum network, but equation is too long and hard to see the message. It would be great if the authors could simplify the equation (14) to reveal the entanglement distribution between four parties. 

4.      The EPS is injected into a single port of star coupler. In general, it is not easy to combine the entangled state into a single path mode. It would be good if the authors could suggest a way to prepare the EPS which has a single path mode.

And there are minor comments as follow.

1.      QWDM (line 44) and WGR (52) are used as an abbreviation without being defined.

Reviewer 3 Report

Comments and Suggestions for Authors

This manuscript presents a theoretical investigation of how to distribute entanglement in a multiuser network using wavelength multiplexing and with minimum resources. Using in particular star couplers to judiciously distribute entangled photons at different wavelengths between users they show that entanglement distribution can performed with a number of wavelength scaling favorably with the number of users as compared to other schemes. They showcase their encoding method using the BBM92 QKD protocol.

The topic of efficient entanglement distribution in quantum networks is a timely and relevant topic for a journal such as Entropy. The concepts introduced in this paper are original and the study is conducted in a convincing manner, so I would give a positive recommendation for publication in Entropy

A minor point that the authors may want to address: the comparison with previous realizations (Sec. 4) is a bit "fluffy" and hard to follow. It could be helpful for the reader to have, on an example with specific coupling values, a clear comparison of the scalings with the number of users of (i) the required wavelengths and (ii) success probability for the different protocols.
